# Electrospun Polycaprolactone/ZnO Nanocomposite Membranes with High Antipathogen Activity

**DOI:** 10.3390/polym14245364

**Published:** 2022-12-08

**Authors:** Elizaveta S. Permyakova, Anton M. Manakhov, Philipp V. Kiryukhantsev-Korneev, Denis V. Leybo, Anton S. Konopatsky, Yulia A. Makarets, Svetlana Yu. Filippovich, Sergey G. Ignatov, Dmitry V. Shtansky

**Affiliations:** 1National University of Science and Technology “MISIS”, 119049 Moscow, Russia; 2Research Institute of Clinical and Experimental Lymphology—Branch of the ICG SB RAS, 2 Timakova st., 630060 Novosibirsk, Russia; 3Bach Institute of Biochemistry, Research Center of Biotechnology, Russian Academy of Sciences, Leninsky Prospect 33, Bld. 2, 119071 Moscow, Russia; 4State Research Center for Applied Microbiology and Biotechnology, 142279 Obolensk, Russia

**Keywords:** polycaprolactone nanofibers, plasma polymerization, ZnO nanoparticles, antibacterial activity, antifungal activity, face masks

## Abstract

The spread of bacterial, fungal, and viral diseases by airborne aerosol flows poses a serious threat to human health, so the development of highly effective antibacterial, antifungal and antiviral filters to protect the respiratory system is in great demand. In this study, we developed ZnO-modified polycaprolactone nanofibers (PCL-ZnO) by treating the nanofiber surface with plasma in a gaseous mixture of Ar/CO_2_/C_2_H_4_ followed by the deposition of ZnO nanoparticles (NPs). The structure and chemical composition of the composite fibers were characterized by SEM, TEM, EDX, FTIR, and XPS methods. We demonstrated high material stability. The mats were tested against Gram-positive and Gram-negative pathogenic bacteria and pathogenic fungi and demonstrated high antibacterial and antifungal activity.

## 1. Introduction

The use of nanofibrous materials has great potential in air filtration applications owing to their unique physical and chemical properties, including high permeability and specific surface, small pore size, and good interconnection. Moreover, they can be functionalized at the nanoscale [1,2]. Nanofibrous filters can capture pathogen-containing droplets and aerosols in an air stream through various mechanisms: inertial impaction, interception, diffusion, gravitational settling, and electrostatic attraction [3]. Due to reduced pore size, ultrafine fibers, and retained surface and volume charges, the use of masks containing nanofiber filters is more effective at filtering dust, bacteria, and viruses than surgical masks and respirators [4,5]. Moreover, the production of nanofibers from biodegradable polymers (PCL, PEO, PLA, etc.) can reduce environmental burden, and since the filtration mechanism is based on structural characteristics, masks can be reused after a simple sterilization process (immersion in ethanol, cleaning in a washing machine). Electrospun nanofiber masks such as Nano Mask (AIRQUEEN, Seoul, South Korea), O_2_ Nano Mask (Viaex Technologies, San Francisco, USA), and FilterLayr™ Eco (NanoLayr, Auckland, New Zealand) are currently commercially available.

Although the polymeric nanofiber network can effectively block various pathogens, there is still a risk of secondary infection caused by the migration of absorbed viruses and/or bacteria in the protective layer. Therefore, the development of self-disinfecting masks is highly desirable. At present, strategies based on the incorporation of ZnO nanoparticles (NPs) inside scaffolds are being actively developed in the field of self-sanitizing materials [6]. The antibacterial properties of ZnO NPs are well known; in addition, recent studies have shown their high antiviral activity against a wide range of viruses (SARS-CoV-2 [7], Chikungunya virus [8], human cytomegalovirus [9], H1N1 influenza virus [10], Hepatitis E/C [11], etc.). Nowadays, there are three mechanisms of antipathogen activity of metal and metal oxide nanoparticles commonly reported in literature, including reactive oxygen species (ROS) generation, ion release, and cell membrane disruption, thereby causing a change in their metabolic activity [12,13]. However, it should be noted that ZnO NPs also bear high photocatalytic properties. Electrons of ZnO NPs can move from the valence band (VB) to the conduction band (CB) due to the absorption of sufficient light energy with the formation of positively charged holes (h+) in VB and free electrons (e−) in CB. [14,15] The electron and holes have strong reductive and oxidative potential, thus under light irradiation, ZnO NPs can react with substances (such as H_2_O and O_2_) in the medium with production of ROS (such as singlet oxygen (1O_2_), hydroxy radical (OH), hydrogen peroxide (H_2_O_2_) and superoxide anion (O_2_−)) [16,17]. Excessive ROS can induce oxidative stress in cells [18].

Although zinc oxide nanoparticles exhibit high antipathogenic activity, it is necessary to ensure their stable binding to the surface to avoid possible cytotoxic effects [19,20]. Thus, it is necessary to develop a method for modifying thin fibers with ZnO NPs that provides a high concentration of ZnO NPs on the filter surface to kill absorbed pathogens and high stability of the obtained materials to prevent toxic effects and loss of antiviral/antibacterial/antifungal activity. In this study, we synthesized polycaprolactone nanofibers (PCL-ref), then deposited a thin polymeric carboxy-containing layer from a CO_2_/C_2_H_4_/Ar gas mixture onto the surface of plasma-modified nanofibers (PCL-COOH), and finally deposited positively charged ZnO NPs (zeta potential +54 mV) on the surface of negatively charged PCL-COOH to ultimately produce PCL ZnO composite. We tested the material stability by immersion in H_2_O for 24 h. Experiments have shown excellent stability of PCL-ZnO composite. The PCL-ZnO filter demonstrated 100% antimicrobial (*Escherichia coli* U20, Staphylococcus aureus CSA154)/antifungal against (*Neurospora crassa*/anti yeast (*Candida parapsilosis* ATCC90018) activity due to incredibly high concentration of ZnO NPs (≥5 at.%).

## 2. Materials and Methods

### 2.1. Synthesis of ZnO NPs

To synthesize ZnO NPs with an average size ~10.5 ± 1.66 nm, 0.305 g (0.073 mol), LiOH was dissolved in 80 mL H_2_O and added dropwise to a Zn(CH_3_COO)_2_ solution (1 g (0.045 mol) Zn(CH_3_COO)_2_*2H_2_O was dissolved in 365 mL of ethylene glycol) at a rate of 80 mL/h. The dispersion was stirred at 5 °C for 2 h. After that, the mixture was heated at 200 °C for 40 min in a microwave oven (Ethnos Easy, Milestone, Italy). After synthesis, ZnO NPs were precipitated by centrifugation followed by washing with H_2_O and drying at room temperature. Particle size was determined from SEM images by calculating the average (>100 ZnO NPs).

### 2.2. Electrospun of PCL Nanofibers

PCL polymer (Sigma Aldrich, Studgart, Germany) with a molecular weight of approximately 80,000 g/mol in an amount of 9 wt.% was dissolved in a 2:1 mixture of acetic acid (99%, Sigma Aldrich, Germany) and formic acid (98%, Sigma Aldrich, Germany). The electrospinning process was carried out on a Nanospider™ NSLAB 500 machine (ELMARCO, Liberec, Czech Republic) using a 20 cm long wire electrode at a voltage of 55 kV according to the protocol described elsewhere [21]. The distance between the high voltage and ground electrodes was 100 mm. Further information on the electrospinning process can be found elsewhere [21]. Spinning PCL samples were denoted as PCL-ref. The thickness of the PCL scaffold was ~100 µm.

### 2.3. COOH Plasma Coating and Deposition of ZnO NPs on Fiber Surface

COOH plasma-polymer layers were deposited using a UVN-2M vacuum system equipped with rotary and oil-diffusion pumps. The plasma was ignited using a Cito 1310-ACNA-N37A-FF radio frequency (RF) power supply (Comet, Flamatt, Switzerland) connected to an RFPG-128 disk generator (Beams & Plasmas, Moscow, Russia) installed in a vacuum chamber. The duty cycle and RF power were set at 5% and 500 W, respectively. The residual pressure in the reactor was below 10^−3^ Pa. CO_2_ (99.995%), Ar (99.998%), and C_2_H_4_ (99.95%) gas flows were set at 50, 16.2, and 6.2 sccm, respectively, and controlled using a Multi-Gas Controller 647C (MKST, Newport, RI, USA). Chamber pressure was measured with a VMB-14 unit (Tokamak Company, Dubna, Russia) and D395-90-000 BOC Edwards controllers. The distance between the RF electrode and the substrate was set to 8 cm. The deposition time was 15 min, which led to the growth of ~100-nm-thick plasma coatings. The plasma-treated PCL nanofibers are referred to as PCL-COOH. To immobilize ZnO NPs on the surface of PCL-COOH fibers, they were immersed in a dispersion of ZnO NPs (C = 2 mg/mL) for 1 h, followed by washing with distilled water.

### 2.4. Characterization

The microstructures of ZnO NPs and surface-modified fine fibers were studied by scanning electron microscopy (SEM) using a JSM-7600F Schottky field emission scanning electron microscope (JEOL Ltd., Tokyo, Japan) equipped with an energy-dispersive X-ray (EDX) detector operated at 15 kV. The samples were coated with a ~40 nm thick Pt layer using a Smart Coater (JEOL Ltd.) to compensate for the surface charge and prevent sample damage. Transmission electron microscopy (TEM) studies of ZnO NPs, including high-resolution TEM and high-angular annual dark field (HAADF) scanning transmission electron microscopy STEM/TEM imaging, were carried out using an FEI Technai Osiris 200 keV S/TEM machine equipped with X-FEG high-brightness electron source. Elemental maps were obtained with an FEI Super-X EDX detection system. The HAADF detector was used for STEM analysis. The chemical and phase compositions were determined by EDX spectroscopy (EDXS) using an 80-mm^2^ X-Max EDX detector (Oxford Instruments, Belfast, UK), Fourier-transform infrared spectroscopy (FTIR) on a Vertex 70v vacuum spectrometer (Bruker, Bremen, Germany), and XPS analysis which was carried out on a PHI VersaProbe III spectrometer (ULVAC-PHI Inc., Osaka, Japan). The XPS spectra were fitted using the CasaXPS software after Shirley-type background subtracting. The binding energy scale was calibrated by shifting the CH_x_ component to 285.0 eV. The size, distribution, and ζ potential of ZnO NPs were determined using a Zetasizer Nano ZS unit (Malvern Panalytical, Malvern, UK). X-ray diffractograms (XRD) were measured on the Difrey-402 instrument (Scientific Instruments, Russia) equipped with position sensitive detector. CrKα radiation (λ = 2.2909 Å) and 300 s exposition time were used during experiments.

### 2.5. Stability Test

A 15 × 15 mm^2^ PCL ZnO sample was immersed in 50 mL of distilled water for 24 h at room temperature. Thereafter, the sample (PCL-ZnO-24 h) was dried and analyzed by SEM, EDXS, and XPS methods as described above.

### 2.6. Antipathogen Activity

To analyze the antibacterial activity/antifungal activity of the samples PCL-ref, PCL-ZnO, two types of hospital bacterial strains (*Staphylococcus aureus* CSA154 and *Escherichia coli* U20), and two types of fungi (*Candida parapsilosis* ATCC90018, *Neurospora crassa*) were used. An overnight culture of the test strains was prepared, suspended in normal saline solution (NS, NaCl = 9 g/L aqueous), decimally titrated, and plated on plates with a solid nutrient media to count colony-forming units (CFUs). The suspension with an approximate concentration of 10^7^ CFU/mL (for the bacterial cultures) and 10^4^ CFU/mL (for the fungal culture) was applied to the surface of each 15*15 mm sample. The samples were incubated at room temperature for 24 h. At the end of incubation, samples were placed in 0.3 mL of NS, pipetted, and the suspension was inoculated to determine the number of CFUs.

## 3. Results

### 3.1. Characterization of ZnO NPs

Figure 1 demonstrates SEM (Figure 1A) and TEM (Figure 1B–D) images of ZnO NPs at various magnifications. Most ZnO NPs are spherical, although some show faceting. The high-resolution TEM image shows that the interplanar spacing is 0.25 nm, which corresponds to the (002) ZnO wurtzite planes [22]. No obvious defects are observed in the NPs. The HAADF-STEM image and corresponding EDXS maps are presented in Figure 1E–G. The presence of zinc and oxygen, which are evenly distributed inside the NPs, indicates the formation of ZnO. Based on 100 measurements using the ImageJ software, the average size of ZnO NPs was determined to be 10.5 ± 1.6 nm, which is in good agreement with the size distribution determined by the DLS method (Figure 1H). The zeta potential of ZnO NPs was +54 mV, which ensures the formation of a stable suspension.

### 3.2. SEM and EDXS Analysis of PCL-ZnO and PCL-ZnO-24 h Samples

The SEM images and the corresponding EDXS elemental maps in samples PCL-ZnO and PCL-ZnO-24 h before and after water immersion are shown in Figure 2. It can be seen that Zn and O are uniformly distributed over the surface of PCL-ZnO and PCL-ZnO-24 h fibers. The sample chemical compositions, determined by the EDXS, before and after the stability test are summarized in Table 1. Soaking in water for 24 h reduces the atomic concentration of Zn element in composition fibers by 0.7 at.%.

### 3.3. XPS Analysis

XPS analysis was used to characterize the sample surface compositions. The atomic compositions of all samples are reported in Table 2. The high Zn concentration (5 at.%) confirms the high dose of loading ZnO NPs into fibers. The amount of Zn did not decrease after soaking in deionized water for 24 h, which confirms the strong adhesion of the ZnO NPs to the nanofibrous matrix. The material compositions differ slightly from the EDXS results (Table 1), which are most likely due to different depths of analysis: ~10 nm (XPS) and ~1000 nm (EDXS). It can also be assumed that not all ZnO NPs were strongly bound to carboxyl (COOH) groups, and some of them were simply mechanically stuck in a three-dimensional fibrous structure and, therefore, were washed away.

To further clarify the surface composition of PCL nanofibers with impregnated ZnO NPs, the high-resolution XPS C1s, O1s, and Zn2p spectra were analyzed. The XPS C1s spectrum of PCL-ref (not shown) was fitted by the sum of three components, namely hydrocarbons CH_x_ (BE = 285 eV), ether group C-O (BE = 286.4 eV), and ester group **C**(O)O (BE = 289.0 eV), which is consistent with previous results [23,24]. The XPS C1s spectrum of ZnO NPs is presented in Figure 3. The presence of minor carbon on the surface indicates that some contaminants were adsorbed on the NPs. The XPS C1s spectra of PCL-ZnO and PCL-ZnO-24 h were fitted by the sum of four components: CH_x_ (BE = 285 eV), ether group C-O (BE = 286.4 eV), carbon doubly bonded to oxygen (C=O, BE = 287.7 eV) and ester group **C**(O)O (BE = 289.0 eV). The full width at half-maximum (FWHM) was set to 1.4 eV for all peaks.

The most important results were obtained from the analysis of the O1s spectra (Figure 3). The XPS spectrum of ZnO NPs mainly consisted of Zn-O contribution centered at 530.2 eV (FWHM = 1.1 eV), which is a typical position for oxygen in metal oxides. The second component at 531.6 eV can be attributed to the bonding of oxygen to carbon that is on the surface due to contamination. The Zn-O peak is also seen in the XPS spectra of samples PCL-ZnO and PCL-ZnO-24 h. The peak area remained unchanged after soaking in water for 25 h (PCL-ZnO-24 h). The Zn2p spectrum also did not change after soaking in water, which indicates only ZnO contribution.

### 3.4. FTIR Spectroscopy Analysis

The FTIR spectra of the studied samples, measured in the range of 4000–400 cm^−1^, are presented in Figure 4. The as-prepared ZnO NPs show peaks at 489, 883, 1413, 1584, and 3440 cm^−1^. The sharp peak positioned at 489 cm^−1^ is attributed to the Zn–O stretching vibrations. Numerous FTIR peaks were observed on all spectra in the range of 1700–600 cm^−1^ can be assigned to C=O, C–O, C-C, and C–H vibrations [25]. A broad maximum observed at 3440 cm^−1^ is ascribed to the stretching vibration of hydroxyl compounds (adsorbed water). The bands at 2946 and 2845 cm^−1^ that are especially noticeable in the FTIR spectra of samples PCL-ref and PCL-ZnO refer to hydrocarbons (CH_3_ and CH_2_ groups, respectively) [26,27]. After PCL-COOH fiber modification with ZnO NPs, a small peak at 489 cm^−1^ is visible (inset), which can be explained by the contribution of ZnO NPs.

### 3.5. XRD Analysis

The results of X-ray diffraction (XRD) analysis of a PCL nanofiber membrane demonstrated in Figure 5, where a sharp crystalline peak at 21.5° (110) and a relatively low-intensity peak at 23.6° (200) were found due to the semi-crystalline nature of PCL polymer [28]. The diffraction peaks located at 31.6°, 34.3°, 36.1°, 47.4°, 56.4°, 62.3°, 63.0°, 68.1°, and 69.2° correspond to the (100), (002), (101), (102), (110), (103), (200), (112) and (201) reflection planes of hexagonal structure of ZnO (JCPDS: 03-065-3411), respectively [29,30]. We detected peaks related to ZnO NPs in sample PCL-ZnO.

### 3.6. Antipathogenic Activity

The antimicrobial activity of samples PCL-ref and PCL-ZnO was studied against Gram-positive (*Staphylococcus aureus* CSA154) and Gram-negative (*Escherichia coli* U20) strains and two fungal strains (*Candida parapsilosis* ATCC90018, *Neurospora crassa*). Colony-forming units are colonies that form on the surface of agar when a sample containing live cells is inoculated on a nutrient medium (Figure 6).

The antibacterial and antifungal potential of PCL-ref and PCL-ZnO was assessed by the number of CFUs after 24 h (Figure 7). In the control sample, the number of CFUs increased by 3–4 orders of magnitude. Compared to the control well without a sample (K), PCL-ref showed noticeable antibacterial activity (approximately 2-log reduction in CFUs). In the case of fungal cultures, the number of CFUs either slightly increased (*Candida parapsilosis*) or decreased (*Neurospora crassa*). The PCL-ZnO sample demonstrated a significant reduction (2-log) in *Candida parapsilosis* (blue column,1) CFUs and 100% antibacterial/antifungal activity against *Neurospora crassa (2)*, *E. coli (3)*, and *S. aureus (4)* strains. The results show that sample PCL ZnO exhibits pronounced antibacterial and antifungal activity against all tested pathogens.

## 4. Discussion

The rapid spread of microorganisms such as bacteria, fungi, and viruses can be extremely dangerous, causing seasonal epidemics or even pandemic situations. Infection can occur via spread through air, water, and contact with infected people, so preventive measures include avoidance of contact, use of chemical disinfectants, UV light sterilization, personal sanitization, and the use of respiratory filters. Such filters should not only block dust and microorganisms but also be self-disinfecting to minimize secondary contamination. Herein, we demonstrate the antibacterial and fungicidal properties of PCL and PCL-ZnO filters. Some recent studies have also shown that PCL fibers exhibit antibacterial activity against *S. aureus*, *P. aeruginosa*, and *E. coli* strains [31,32,33]. This can be explained by the hydrophobic properties of PCL fibers [34], which were also demonstrated in our previous work [35].

Ag [36,37] and Cu [38,39,40] NPs are widely used along with ZnO NPs to impart antibacterial, antifungal, and antiviral properties to nanofiber filters. However, the utilization of ZnO NPs as an antimicrobial agent is more cost-effective, and, in addition, the concept of creating self-cleaning ZnO-based materials is highly promising due to their photocatalytic properties [41].

Using antibacterial metal and metal oxide nanoparticles is a highly effective tool in the fight against pathogenic microorganisms, especially with multidrug resistance [42]. The composite nanofibers loaded by antibacterial NPs can be applied in different fields including wound healing [43,44,45], water [46,47], and air filtration [38,48]. Hiremath et al. [48] prepared electrospun nanofibers based on PVP:TiO2_10%_ matrix doped by silver in different concentrations (0.2, 0.4, 0.6, and 0.8 wt.%). The antipathogenic tests demonstrated a gradual reduction in several viable organisms (bacterial species: Bacillus subtilis (BS), Staphylococcus epidermidis (SE), *Staphylococcus aureus* (SA), and Staphylococcus faecalis (SF); fungal strains: Candida tropicalis (CT), *Candida parapsilosis* (CP), Candida albicans (CA) and Candida glabrata (CG)) with increasing concentration of Ag. Alharbi et al. [49] demonstrated high antibiofilm activity core-shell coaxial nanofibers PVA/PLA (AgNO_3_) against two prokaryotes (Pseudomonas aeruginosa and *Staphylococcus aureus*) and a eukaryote (Candida albicans). ZnO-modified gelatin nanofibers demonstrated high antibacterial activity against Gram-positive (Staphyloccocus aureus and Bacillus pumilus) and Gram-negative (*Escherichia coli* and Pseudomonas fluorescens) strains. However, it should be noted that the production of ZnO NPs based on simple methods and is more cost-effective than the synthesis of AgNPs [50].

## 5. Conclusions

A ZnO nanoparticle-modified PCL fibrous material has been obtained with great potential for use in respiratory protection. The inherently flexible platform based on electrospinning fibers decorated with ZnO nanoparticles exhibits enhanced antibacterial and fungicidal activity. The obtained results of the EDXS and XPS analysis demonstrate the high stability of the obtained PCL-ZnO composites. This work shows the promise of PCL fibers in various biomedical applications, including in a protective filter layer.

## Figures and Tables

**Figure 1 polymers-14-05364-f001:**
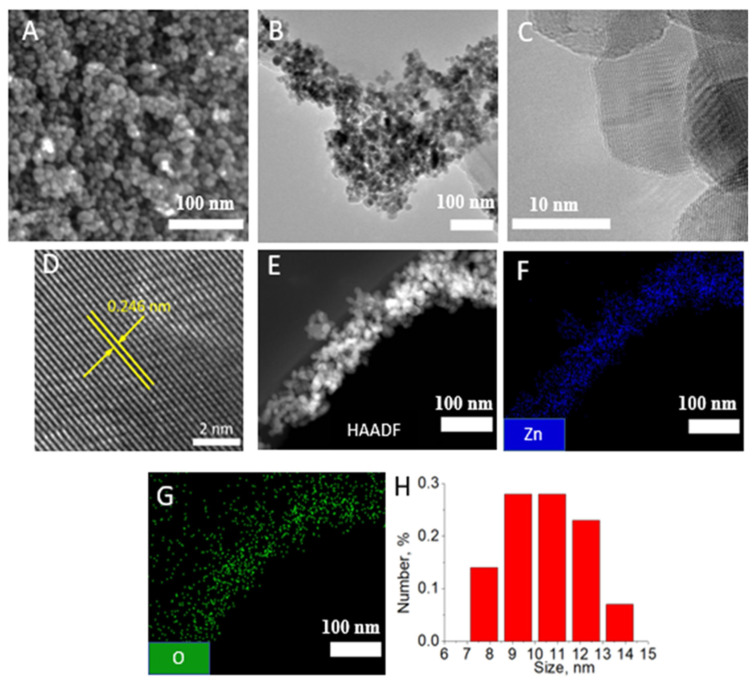
SEM (**A**), TEM (**B**), and high-resolution TEM (**C**,**D**) images of ZnO NPs. HAADF-STEM image (**E**) with corresponding EDXS elemental maps (**F**,**G**). Size distribution of ZnO NPs estimated by DLS method (**H**).

**Figure 2 polymers-14-05364-f002:**
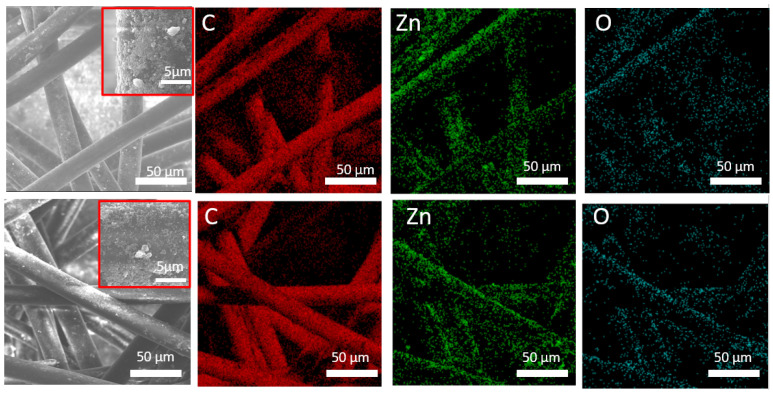
SEM images and corresponding EDXS elemental maps of PCL-ZnO and PCL-ZnO-24 h samples.

**Figure 3 polymers-14-05364-f003:**
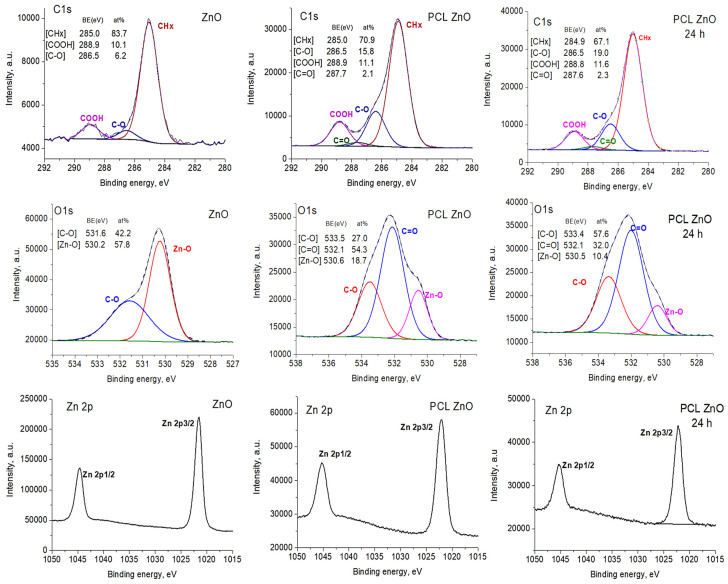
XPS C1s, O1s, and Zn2p spectra of samples ZnO NPs, PCL-ZnO and PCL-ZnO-24 h.

**Figure 4 polymers-14-05364-f004:**
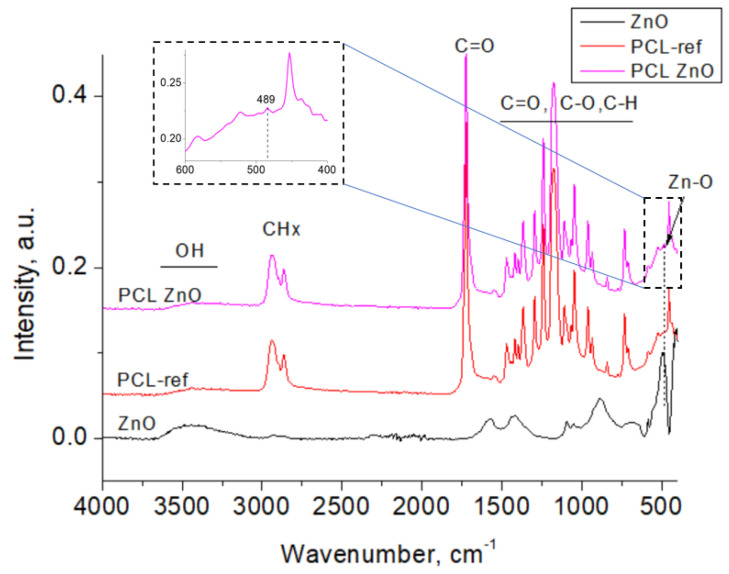
FTIR spectra of samples ZnO NPs, PCL-ref, and PCL-ZnO.

**Figure 5 polymers-14-05364-f005:**
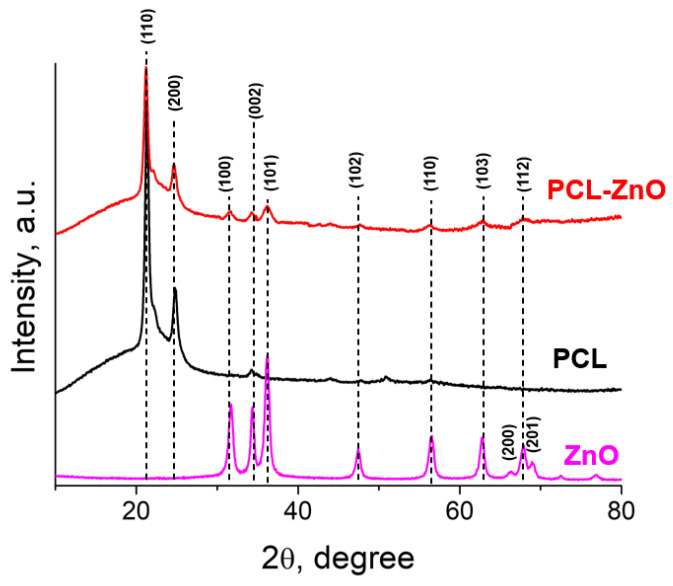
XRD pattern of ZnO NPs, PCL and PCL-ZnO membrane.

**Figure 6 polymers-14-05364-f006:**
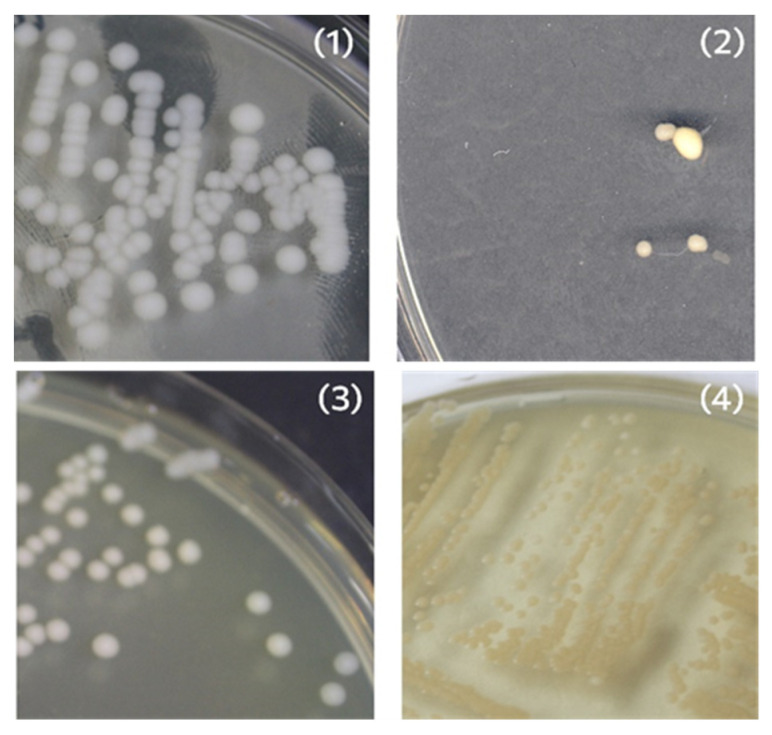
Bacterial colony-forming units *Candida parapsilosis* ATCC90018 (**1**), *Neurospora crassa* (**2**), *Escherichia coli* U20 (**3**) and *Staphylococcus aureus* CSA154 (**4**).

**Figure 7 polymers-14-05364-f007:**
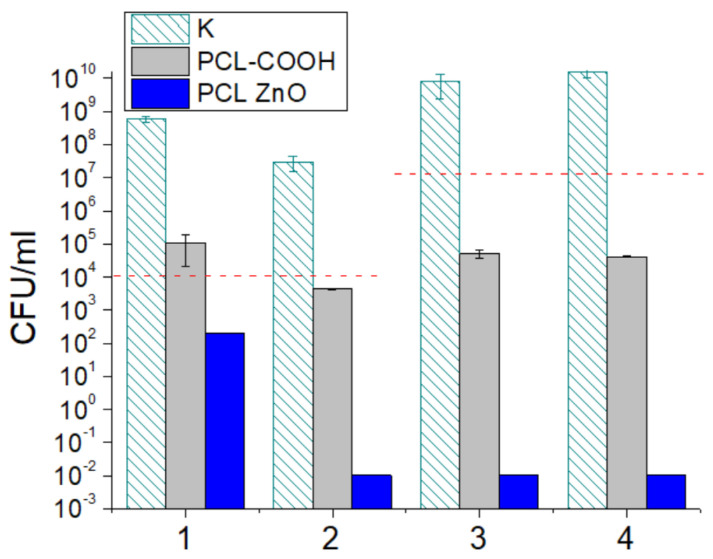
Antifungal and antibacterial activities of PCL-COOH and PCL-ZnO against *Candida parapsilosis* ATCC90018 (1), *Neurospora crassa* (2), *Escherichia coli* U20 (3) and *Staphylococcus aureus* CSA154 (4) assessed by CFU count after 24 h. K—control without sample. Dashed horizontal lines show initial cell concentrations.

**Table 1 polymers-14-05364-t001:** Sample atomic compositions measured by EDXS.

Sample	C, at.%	O, at.%	Zn, at.%	Pt, at.%
PCL-ZnO	86.9	6.0	5.7	1.4
PCL-ZnO-24 h	88.4	5.3	5.0	1.3

**Table 2 polymers-14-05364-t002:** Sample atomic compositions measured by XPS.

Sample	[C], at.%	[O], at.%	[Zn], at.%
PCL-ref	73.9	26.1	0.0
ZnO (powder)	17.4	42.3	40.3
PCL-ZnO	69.0	25.9	5.0
PCL-ZnO-24 h	70.8	24.1	5.1

## Data Availability

Data is available from the corresponding author upon a reasonable request.

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
