# Peer review of "Electrospun Polycaprolactone/ZnO Nanocomposite Membranes with High Antipathogen Activity"

_polymers, 2022, doi:10.3390/polym14245364_

Round 1

Reviewer 1 Report

This work reports the preparation of polycaprolactone nanofiber membrane coated with ZnO nanoparticles and their antimicrobial properties. The intention for the research is interesting and worth of studying. Nevertheless, additional experimental data should be included in conjunction with the improvement of manuscript writing to meet the high criteria for publication in a good journal such as Polymers. Following are the reviewer’s concerns:

1)     The authors do not show any experimental results involved the use of the developed membranes as filters for respiratory face masks. Therefore, the title should be revised based on the content presented in the main text, rather than telling their perspective.

2)     In the introduction and discussion part, possible mechanisms for the microbicidal activity of the ZnO NPs or ZnO-PCL membranes should be provided.

3)     The term “plasma-coated PCL nanofibers” should be revised. Plasma is a state of matter, which cannot be coated on the nanofibers. Terms such as “plasma-treated nanofibers” can be considered. Moreover, how the authors know the plasma coating has thickness of 100 nm? If there is no evidence (e.g., microscopic images), it is better avoiding this specification.

4)     Heading for section 2.6 should be revised as “Antipathogenic activity”.

5)     XRD patterns of ZnO NPs and/or ZnO-PCL should be included to provide a solid evidence for the formation of the membrane.

6)      A couple of references for the characteristic peaks of hydrocarbons in the FTIR spectra should be provided.

7)     In figure 5: why the final product PCL-ZnO membrane was tested only with Candida parapsilosis? How about other types?

8)     Photos of bacterial cultures should be provided.

9)     A comparison with other antimicrobial nanoparticles such as Ag NPs (https://doi.org/10.1016/j.msec.2020.111497; https://doi.org/10.1016/j.colsurfb.2021.111856 ) should be discussed. Is it possible to replace ZnO NPs with Ag NPs?     

Author Response

Dear Referee,

We are thankful for your comments and suggestions that helped to improve our work. We conducted the new experiments and added to the paper. answers to all your comments are below and in the attached PDF for your convenience

Reviewer 1

Q1:The authors do not show any experimental results involved the use of the developed membranes as filters for respiratory face masks. Therefore, the title should be revised based on the content presented in the main text, rather than telling their perspective.

 Answer: The title of the manuscript was changed:

Polycaprolactone Nanofiber Membrane with ZnO Nanoparticles as Perspective Antipathogen Filters

Q2: In the introduction and discussion part, possible mechanisms for the microbicidal activity of the ZnO NPs or ZnO-PCL membranes should be provided.

 Answer: The next paragraph was added in the Introduction section

The antipathogenic mechanisms of ZnO nanoparticles included the formation of reactive oxygen species (ROS), the interaction of nanoparticles with bacteria, subsequently damaging the bacterial cell, and the release of Zn2+ ions. (1)

  1. Shi LE, Li ZH, Zheng W, Zhao YF, Jin YF, Tang ZX. Synthesis, antibacterial activity, antibacterial mechanism and food applications of ZnO nanoparticles: A review. Food Addit Contam - Part A 2014;31(2):173–86. Available from: http://dx.doi.org/10.1080/19440049.2013.865147

Q3:The term “plasma-coated PCL nanofibers” should be revised. Plasma is a state of matter, which cannot be coated on the nanofibers. Terms such as “plasma-treated nanofibers” can be considered. Moreover, how the authors know the plasma coating has thickness of 100 nm? If there is no evidence (e.g., microscopic images), it is better avoiding this specification.

 Answer: The term “plasma-coated PCL nanofibers” was changed to “plasma-treated nanofibers” in the text of the article.

Q4: Heading for section 2.6 should be revised as “Antipathogenic activity”.

 Answer: Heading for section 2.6 was changed to “Antipathogenic activity”

Q5: XRD patterns of ZnO NPs and/or ZnO-PCL should be included to provide solid evidence for the formation of the membrane.

Answer: The next paragraphs and Figure were added

In Materials and Methods section:

X-Ray diffractograms (XRD) were measured on Difrey-402 instrument (Scientific Instruments, Russia) equipped with position sensitive detector. CrKa radiation (λ = 2.2909Å) and 300 s exposition time was used during experiments.

In Results section: The results of X-ray diffraction (XRD) analysis of a PCL nanofiber membrane demonstrated in Figure5, where a sharp crystalline peak at 21.5°(110) and a relatively low-intensity peak at 23.6° (200) were found due to the semi-crystalline nature of PCL polymer.(2) However we didn’t detect peaks of ZnO, which can be explained by the small size of NPs

  1. Kao HH, Kuo CY, Tagadur Govindaraju D, Chen KS, Chen JP. Polycaprolactone/Chitosan Composite Nanofiber Membrane as a Preferred Scaffold for the Culture of Mesothelial Cells and the Repair of Damaged Mesothelium. Int J Mol Sci. 2022;23(17).

Q6: A couple of references for the characteristic peaks of hydrocarbons in the FTIR spectra should be provided.

 The next references were added

refer to hydrocarbons (CH3 and CH2 groups, respectively)(3,4)

  1. Wang Y, Wei W, Zhang Y, Hanson RK. A new strategy of characterizing hydrocarbon fuels using FTIR spectra and generalized linear model with grouped-Lasso regularization. Fuel [Internet]. 2021;287(July):119419. Available from: https://doi.org/10.1016/j.fuel.2020.119419
  2. Sakaguti KY, Wang SH. Preparation of poly(3-hydroxybutyrate-b-∈-caprolactone) by reactive extrusion and production of electrospun fibrous mats. J Braz Chem Soc. 2021;32(2):355–62.

Q7: In figure 5: why the final product PCL-ZnO membrane was tested only with Candida parapsilosis? How about other types?

Answer: We tested PCL and PCL-ZnO membrane against Gram-positive (Staphylococcus aureus CSA154) and Gram-negative (Escherichia coli U20) strains and two fungal strains (Candida parapsilosis ATCC90018, Neurospora crassa). We rewritten the paragraph to clarify this:

“The antimicrobial activity of samples PCL-ref and PCL ZnO was studied against Gram-positive (Staphylococcus aureus CSA154) and Gram-negative (Escherichia coli U20) strains and two fungal strains (Candida parapsilosis ATCC90018 , Neurospora crassa). The antibacterial and antifungal potential of PCL-ref and PCL-ZnO was assessed by the number of CFUs after 24 h (Fig. 5). In the control sample, the number of CFUs increases by 3-4 orders of magnitude. Compared to the control well without a sample (K), PCL-ref showed noticeable antibacterial activity (approximately 2-log reduction in CFUs). In the case of fungal cultures, the number of CFUs either slightly increased (Candida parapsilosis) or decreased (Neurospora crassa). The PCL-ZnO sample demonstrated a significant reduction (2-log) in Candida parapsilosis CFUs ((blue column,1) and 100% antibacterial/antifungal activity against Neurospora crassa (2), E. coli(3), and S. aureus(4) strains. In this case, there is no blue columns in 2, 3 and 4 because all microorganisms have been eliminated by PCL-ZnO and colony-forming units are not detected (Figure 5.). “

Q8: Photos of bacterial cultures should be provided.

Answer: We added next sentence and a new Figure

Colony-forming units are colonies that form on the surface of agar when a sample containing live cells is inoculated on a nutrient medium (Fig. 6).

Figure 6. Bacterial colony forming units (1), Fungal colony forming units (2)

Q9: A comparison with other antimicrobial nanoparticles such as Ag NPs (https://doi.org/10.1016/j.msec.2020.111497; https://doi.org/10.1016/j.colsurfb.2021.111856 ) should be discussed. Is it possible to replace ZnO NPs with Ag NPs?  

Reviewer 2 Report

In this manuscript, the authors developed ZnO-modified polycaprolactone nanofibers (PCL-ZnO) by treating the nanofiber surface with plasma in a gaseous mixture of Ar/CO2/C2H4 followed by the deposition of ZnO nanoparticles. The structure and chemical composition of the composite fibers were characterized by many techniques. The obtained results of the EDXS and XPS analysis demonstrate the high stability of the PCL-ZnO composites. ZnO nanoparticles exhibit enhanced antibacterial and fungicidal activity. This study is good and important to prepare protective filter layers from polycaprolactone which can be sued in various biomedical applications. The interpretations of the results are well discussed. The quantity and quality of the figures are appropriate. We believe that this research subject is promising for developing new nanofiber composites for biomedical applications.

Summary: I recommend publishing this manuscript after considering my comments on the attached file.

Author Response

Dear Referee,

We are grateful for your comments and suggestions provided in the attached PDF file.

All problems were resolved in the revised  version.

We have made an extensive revision and proof reading.

Round 2

Reviewer 1 Report

The authors have tried to include several experimental data, however, most of them still need to be improved by either experiments or cautious discussion with references. Please see the following comments from the reviewer for your consideration to elaborate the manuscript.

1) This is a research article, so that the title must reflect the content of the manuscript rather than containing its perspective uses. Please revise again.

2) The discussion/introduction "The antipathogenic mechanisms of ZnO nanoparticles included the formation of reactive oxygen species (ROS), the interaction of nanoparticles with bacteria, subsequently damaging the bacterial cell, and the release of Zn2+ ions. (1)" is too vague. Please note that not all published papers are entirely correct and be cautious when quoting/citing the content in the reference.

What is the formation of reactive oxygen species (ROS)? it comes from ZnO NPs? or from microorganism? or others? Basically, NPs can physically interact with microorganisms to damage their structure, thus causing changing in their metabolic activities and ROS or directly destroy/kill bacteria/fungi. Another route is the release of metal ions, which infiltrate into the microorganisms and induce changes in their metabolic activities (and cause ROS). This information can be found in the suggested literature [(https://doi.org/10.1016/j.msec.2020.111497; https://doi.org/10.1016/j.colsurfb.2021.111856 )]. 

3) It is very careless to make a statement that XRD characteristics of ZnO NPs cannot be detected due to their small sizes. Please note that it takes 2 seconds to google search and find that hundreds of XRD peaks of ZnO NPs composited with polymers. Please do measurement again and find a better experimental set up to obtain the expect results. If cannot, the reviewer can provide hints in next possible rounds.

4) Figure 5 (both original and revised) still shows the final product PCL-ZnO membrane tested only with Candida parapsilosis? How about other types? Please again to address this comment directly.

5) Please include the photos of all test microorganisms (not only two types) and these data must be presented prior to the original figure 5 or revised figure 6.

6) If the authors want to develop commercial products or even research products, the authors should compare the antibacterial/fungal activities of ZnO NPs with others such as Ag NPs. Thereby audience can have a better evaluation and broader view if they want to commericiallize the products. 

Author Response

First of all, we want to express our sincere gratefulness to the Reviewer for his/her great comments and suggestions. We have made all necessary additionally requested experiments, analyzed recommended literature and expanded the discussion. Please find our answers below and in the attached file.

The authors have tried to include several experimental data, however, most of them still need to be improved by either experiments or cautious discussion with references. Please see the following comments from the reviewer for your consideration to elaborate the manuscript.

Q1. This is a research article, so that the title must reflect the content of the manuscript rather than containing its perspective uses. Please revise again.

Answer: The title was changed

Electrospun polycaprolactone/ZnO nanocomposite membranes with high antipathogen activity

2) The discussion/introduction "The antipathogenic mechanisms of ZnO nanoparticles included the formation of reactive oxygen species (ROS), the interaction of nanoparticles with bacteria, subsequently damaging the bacterial cell, and the release of Zn2+ ions. (1)" is too vague. Please note that not all published papers are entirely correct and be cautious when quoting/citing the content in the reference.

What is the formation of reactive oxygen species (ROS)? it comes from ZnO NPs? or from microorganism? or others? Basically, NPs can physically interact with microorganisms to damage their structure, thus causing changing in their metabolic activities and ROS or directly destroy/kill bacteria/fungi. Another route is the release of metal ions, which infiltrate into the microorganisms and induce changes in their metabolic activities (and cause ROS). This information can be found in the suggested literature [(https://doi.org/10.1016/j.msec.2020.111497; https://doi.org/10.1016/j.colsurfb.2021.111856 )]. 

Answer: We do understand the concerns from the reviewer and his suggestion to critically review the information provided in references. However, the review paper which we cited (DOI: 10.1080/19440049.2013.865147) was published in Q1 Journal and was cited 269 times, so we do belive that it is reasonable to rely on the refered data.Nevetheless we did expand our manuscript and introiduced following paragraph to our text.

The next paragraph was added in Introduction part

Nowadays, there are three mechanisms of antipathogen activity of metal and metal oxide nanoparticles commonly reported in literatures, included reactive oxygen species (ROS) generation, ions release and cell membrane disruption thereby causing a change in their metabolic activity. (1,2) However it should be noted that ZnO NPs have high photocatalytic properties. Electrons of ZnO NPs can move from valence band (VB) to conduction band (CB) due to absorbing sufficient light energy with the formation of positively-charged holes (h+ ) in VB and free electrons (e- ) in CB. (3,4)The electron and holes have strong reductive and oxidative potential, thus under light irradiation ZnO NPs can react with substances (such as H2O and O2) in the medium with production ROS (such as single oxygen (1O2), hydroxy radical (∙OH), hydrogen peroxide (H2O2) and superoxide anion (∙O2 − )). (5,6)The excessive ROS can induce oxidative stress in cells. (7)

  1. Nguyen DD, Luo LJ, Lai JY. Toward understanding the purely geometric effects of silver nanoparticles on potential application as ocular therapeutics via treatment of bacterial keratitis. Mater Sci Eng C [Internet]. 2021;119(February 2020):111497. Available from: https://doi.org/10.1016/j.msec.2020.111497
  2. Nguyen DD, Lue SJ, Lai JY. Tailoring therapeutic properties of silver nanoparticles for effective bacterial keratitis treatment. Colloids Surfaces B Biointerfaces [Internet]. 2021;205(February):111856. Available from: https://doi.org/10.1016/j.colsurfb.2021.111856
  3. Du M, Zhao W, Ma R, Xu H, zhu Y, Shan C, et al. Visible-light-driven photocatalytic inactivation of S. aureus in aqueous environment by hydrophilic zinc oxide (ZnO) nanoparticles based on the interfacial electron transfer in S. aureus/ZnO composites. J Hazard Mater [Internet]. 2021;418:126013. Available from: https://doi.org/10.1016/j.jhazmat.2021.126013
  4. Chang J Sen, Strunk J, Chong MN, Poh PE, Ocon JD. Multi-dimensional zinc oxide (ZnO) nanoarchitectures as efficient photocatalysts: What is the fundamental factor that determines photoactivity in ZnO? J Hazard Mater [Internet]. 2020;381(June 2019):120958. Available from: https://doi.org/10.1016/j.jhazmat.2019.120958
  5. Adhikari S, Banerjee A, Eswar NKR, Sarkar D, Madras G. Photocatalytic inactivation of E. Coli by ZnO-Ag nanoparticles under solar radiation. RSC Adv. 2015;5(63):51067–77.
  6. Sirelkhatim A, Mahmud S, Seeni A, Kaus NHM, Ann LC, Bakhori SKM, et al. Review on zinc oxide nanoparticles: Antibacterial activity and toxicity mechanism. Nano-Micro Lett [Internet]. 2015;7(3):219–42. Available from: http://dx.doi.org/10.1007/s40820-015-0040-x
  7. Tiwari V, Mishra N, Gadani K, Solanki PS, Shah NA, Tiwari M. Mechanism of anti-bacterial activity of zinc oxide nanoparticle against Carbapenem-Resistant Acinetobacter baumannii. Front Microbiol. 2018;9(JUN):1–10.
  8. Saha RK, Debanath MK, Saikia E. Multifractal analysis of ZnO nanoparticles. Mater Sci Eng C [Internet]. 2020;106:110177. Available from: https://doi.org/10.1016/j.msec.2019.110177
  9. Álvarez-Chimal R, García-Pérez VI, Álvarez-Pérez MA, Arenas-Alatorre JÁ. Green synthesis of ZnO nanoparticles using a Dysphania ambrosioides extract. Structural characterization and antibacterial properties. Mater Sci Eng C [Internet]. 2021;118:111540. Available from: https://doi.org/10.1016/j.msec.2020.111540

3) It is very careless to make a statement that XRD characteristics of ZnO NPs cannot be detected due to their small sizes. Please note that it takes 2 seconds to google search and find that hundreds of XRD peaks of ZnO NPs composited with polymers. Please do measurement again and find a better experimental set up to obtain the expect results. If cannot, the reviewer can provide hints in next possible rounds.

Answer: We do apologize for our unsufficient quality of previous XRD results. We have applied longer exposure time and better adjusted the acquisition conditions. We increased time of measurements and detected peaks related to ZnO NPs. The next paragraph was added

The diffraction peaks located at 31.6°, 34.3°, 36.1°, 47.4°, 56.4°, 62.3°, 63.0â—¦, 68.1â—¦, and 69.2â—¦ correspond to the (100), (002), (101), (102), (110), (103), (200), (112) and (201) reflection planes of hexagonal structure of ZnO (JCPDS: 03-065-3411), respectively. (8,9) We detected peaks related to ZnO NPs in sample PCL-ZnO.

Figure 5. XRD pattern of ZnO NPs, PCL and PCL-ZnO membrane.

4) Figure 5 (both original and revised) still shows the final product PCL-ZnO membrane tested only with Candida parapsilosis? How about other types? Please again to address this comment directly.

Answer: Since our previous description was not sufficient, we have improved the description in our text (changes are highlighted). Hopefully, now all indications are clear.

“The PCL-ZnO sample demonstrated a significant reduction (2-log) in Candida parapsilosis CFUs ((blue column,1) and 100% antibacterial/antifungal activity against Neurospora crassa (2), E. coli(3), and S. aureus(4) strains. In this case, there is no blue columns in 2, 3 and 4 because all microorganisms have been eliminated by PCL-ZnO and colony-forming units are not detected (Figure 5.) “

We increased the Y scale (CFU counts) in the side minimal values to demonstrate that Neurospora crassa, E. coli, and S. aureus strains were eliminated totally on the surface of PCL ZnO samples. Since we have LOG scale, we couldn’t demonstrate ZERO, so we paste 0.01 value (CFU counts) in PCL ZnO column to demonstrate that we tested PCL ZnO samples against all 4 pathogens. The numbers under the columns corresponds Candida parapsilosis (1), Neurospora crassa (2), E. coli(3), and S. aureus(4) strains, how it is written in the description of Figure.

5) Please include the photos of all test microorganisms (not only two types) and these data must be presented prior to the original figure 5 or revised figure 6.

Answer: The photos of all test microorganisms presented on Figure 6

Figure 6. Bacterial colony forming units Candida parapsilosis ATCC90018 (1), Neurospora crassa (2), Escherichia coli U20 (3) and Staphylococcus aureus CSA154 (4)

6) If the authors want to develop commercial products or even research products, the authors should compare the antibacterial/fungal activities of ZnO NPs with others such as Ag NPs. Thereby audience can have a better evaluation and broader view if they want to commericiallize the products. 

Answer: The next paragraph was added in Discussion chapter

Using antibacterial metal and metal oxide nanoparticles is a highly effective tool in the fight against pathogenic microorganisms, especially with multidrug resistance. [1] The composite nanofibers loaded by antibacterial NPs can be applied in different fields including wound healing[2–4], water[5,6], and air filtration[7,8]. Hiremath et al. [6] prepared electrospun nanofibers based on PVP:TiO210% matrix doped by silver in different concentrations(0.2, 0.4, 0.6, and 0.8 wt%). The antipathogenic tests demonstrated a gradual reduction in several viable organisms (bacterial species: Bacillus subtilis (BS), Staphylococcus epidermidis (SE), Staphylococcus aureus (SA), and Staphylococcus faecalis (SF); fungal strains: Candida tropicalis(CT), Candida parapsilosis (CP), Candida albicans (CA) and Candida glabrata (CG)) with increasing of concentration Ag. Alharbi et al.[9] demonstrated high antibiofilm activity core-shell coaxial nanofibers PVA/PLA(AgNO3) against two prokaryotes (Pseudomonas aeruginosa and Staphylococcus aureus) and a eukaryote (Candida albicans). ZnO-modified gelatin nanofibers demonstrated high antibacterial activity against Gram-positive (Staphyloccocus aureus and Bacillus pumilus) and Gram-negative (Escherichia coli and Pseudomonas fluorescens) strains. However it should be noted that the production of ZnO NPs based on simple methods and is more cost-effective than the synthesis of AgNPs. [10]

  1. Kheiri, S.; Liu, X.; Thompson, M. Nanoparticles at biointerfaces: Antibacterial activity and nanotoxicology. Colloids Surfaces B Biointerfaces 2019, 184, 110550.
  2. Ciloglu, N.S.; Mert, A.I.; Doʇan, Z.; Demir, A.; Cevan, S.; Aksaray, S.; Tercan, M. Efficacy of silver-loaded nanofiber dressings in Candida albicans-contaminated full-skin thickness rat burn wounds. J. Burn Care Res. 2014, 35, e317–e320.
  3. Sitnikova, N.A.; Solovieva, A.O.; Permyakova, E.S.; Sheveyko, A.N.; Shtansky, D. V.; Manakhov, A.M. Silver Ions Incorporation into Nanofibers for Enhanced hMSC Viability. Chem. 2022, 4, 931–939.
  4. He, C.; Liu, X.; Zhou, Z.; Liu, N.; Ning, X.; Miao, Y.; Long, Y.; Wu, T.; Leng, X. Harnessing biocompatible nanofibers and silver nanoparticles for wound healing: Sandwich wound dressing versus commercial silver sulfadiazine dressing. Mater. Sci. Eng. C 2021, 128, 112342.
  5. Cheng, Z.; Zhao, S.; Han, L. A novel preparation method for ZnO/γ-Al2O3 nanofibers with enhanced absorbability and improved photocatalytic water-treatment performance by Ag nanoparticles. Nanoscale 2018, 10, 6892–6899.
  6. Hiremath, L.; Kumar, N.S.; Gupta, P.K.; Srivastava, A.K.; Choudhary, S.; Suresh, R.; Keshamma, E. Synthesis, characterization of TiO2 doped nanofibres and investigation on their antimicrobial property. J. Pure Appl. Microbiol. 2019, 13, 2129–2140.
  7. Manakhov, A.M.; Permyakova, E.S.; Sitnikova, N.A.; Tsygankova, A.R.; Alekseev, A.Y.; Solomatina, M. V.; Baidyshev, V.S.; Popov, Z.I.; Blahová, L.; Eliáš, M.; et al. Biodegradable Nanohybrid Materials as Candidates for Self-Sanitizing Filters Aimed at Protection from SARS-CoV-2 in Public Areas. Molecules 2022, 27, 1333.
  8. De Sio, L.; Ding, B.; Focsan, M.; Kogermann, K.; Pascoal-Faria, P.; Petronela, F.; Mitchell, G.; Zussman, E.; Pierini, F. Personalized Reusable Face Masks with Smart Nano-Assisted Destruction of Pathogens for COVID-19: A Visionary Road. Chem. - A Eur. J. 2021, 27, 6112–6130.
  9. Alharbi, H.F.; Luqman, M.; Khan, S.T. Antibiofilm activity of synthesized electrospun core-shell nanofiber composites of PLA and PVA with silver nanoparticles. Nanotechnology 2018, 29.
  10. Huaxu, L.; Fuqiang, W.; Dong, Z.; Ziming, C.; Chuanxin, Z.; Bo, L.; Huijin, X. Experimental investigation of cost-effective ZnO nanofluid based spectral splitting CPV/T system. Energy 2020, 194, 116913.

Round 3

Reviewer 1 Report

The authors have significantly improved the paper quality by further experiments and discussion. Therefore, the reviewer think it is worth publishing the work in Polymers journal.